# Preparation and Characterization of a Hypoglycemic Complex of Gallic Acid–Antarctic Krill Polypeptide Based on Polylactic Acid–Hydroxyacetic Acid (PLGA) and High-Pressure Microjet Microencapsulation

**DOI:** 10.3390/foods13081177

**Published:** 2024-04-12

**Authors:** Mengjie Li, Puyu Chen, Yichen Lin, Song Miao, Hairong Bao

**Affiliations:** 1College of Food Science & Technology, Shanghai Ocean University, Shanghai 201306, China; 2Teagasc Food Research Centre, Moorepark, Fermoy, Co., P61 C996 Cork, Ireland

**Keywords:** microemulsion, high-pressure microjet, microcapsule, hyperglycemic, GA-AKP, delayed release

## Abstract

Gallic acid–Antarctic krill peptides (GA-AKP) nanocapsules (GA-AKP-Ns) were prepared using a dual delivery system with complex emulsion as the technical method, a high-pressure microjet as the technical means, polylactic acid–hydroxyacetic acid (PLGA) as the drug delivery vehicle, and GA-AKP as the raw material for delivery. This study aimed to investigate the effects of microjet treatment and the concentration of PLGA on the physicochemical properties and stability of the emulsion. Under optimal conditions, the physicochemical properties and hypoglycemic function of nano-microcapsules prepared after lyophilization by the solvent evaporation method were analyzed. Through the microjet treatment, the particle size of the emulsion was reduced, the stability of the emulsion was improved, and the encapsulation rate of GA-AKP was increased. The PLGA at low concentrations decreased the particle size of the emulsion, while PLGA at high concentrations enhanced the encapsulation efficiency of the emulsion. Additionally, favorable results were obtained for emulsion preparation through high-pressure microjet treatment. After three treatment cycles with a PLGA concentration of 20 mg/mL and a microjet pressure of 150 MPa (manometric pressure), the emulsion displayed the smallest particle size (285.1 ± 3.0 nm), the highest encapsulation rates of GA (71.5%) and AKP (85.2%), and optimal physical stability. GA-AKP was uniformly embedded in capsules, which can be slowly released in in vitro environments, and effectively inhibited α-amylase, α-glucosidase, and DPP-IV at different storage temperatures. This study demonstrated that PLGA as a carrier combined with microjet technology can produce excellent microcapsules, especially nano-microcapsules, and these microcapsules effectively improve the bioavailability and effectiveness of bioactive ingredients.

## 1. Introduction

Nanocapsules have emerged as a popular research topic in bioactive ingredients and drug delivery, attracting global attention. The preparation technology for nanocapsules is rapidly advancing and gaining interest. However, the conventional spray-drying process, which involves continuous high temperatures, may pose a potential threat to the physicochemical properties of the raw materials. Although alternative methods for encapsulating bioactive substances, such as layer-by-layer self-assembly (LBL), fermentation cell methods, and ionic gel approaches, have benefits in reducing the impact on the functionality of raw materials, they are limited by high operational requirements and low preparation efficiency. In contrast, the solvent evaporation method, involving pre-preparation of core and wall materials into emulsions or compound emulsions, followed by obtaining nanocapsules through freeze-drying, not only avoids the damage to raw materials at high temperatures but also has the advantages of efficient, easy operation, and a high encapsulation rate. In this method, the primary emulsion of core and wall material is emulsified by the high-pressure microjet method. Physical treatment technology is an excellent method for preparing microemulsions, capable of producing uniform droplets at the nanoscale [1]. Significantly, it effectively addresses concerns related to drug release caused by poor uniformity of emulsion particle size and diverse droplet morphology, contributing to a reduction in dosage frequency. The homogenization pressure and treatment times are the main factors affecting the physical and chemical properties of emulsion [2], but few studies have been reported.

Most of the currently used drug delivery vehicles in clinical use are viral derivatives, which carry the risk of damaging human genes [3]. An alternative to these viral derivatives is polylactic acid–hydroxyacetic acid (PLGA), a safe and efficient polymeric material with variants such as PLGA-75:25, PLGA-50:50, and PLGA-25:75 [4]. The final metabolites of PLGA in vivo are water and carbon dioxide. PLGA has been regarded as a reliable carrier for protein and peptide drugs. Additionally, it has been used as an alternative to oral insulin [5]. There have been studies on the delivery of PLGA loaded with polyphenols [4], but dual delivery systems with PLGA loaded with proteins and phenolics have rarely been reported [6].

At present, traditional drug delivery methods for diabetes require frequent oral doses or direct insulin injections, which can cause physical discomfort for patients. Delivering hypoglycemic natural ingredients in the form of particles can reduce the side effects of drug overdose on the body, and improve pharmacokinetics through sustained release, ultimately improving patient convenience [7,8,9]. However, the simple structure of particles has limited their application as multi-component loading systems, hindering the synergistic effects of components. Therefore, it is of practical interest to explore a particle that can deliver multiple ingredients simultaneously.

Encapsulation technology not only supports the delivery of small molecules and stability in vivo but also aims to protect the active ingredient during storage. The capsule shell isolates the active ingredient from the external environment, preventing loss of raw materials, prolonging the half-life, and diminishing the risks of evaporation and volatile degradation [10,11,12]. Microencapsulation is a method in which tiny particles or droplets are surrounded by a coating wall, or embedded in a homogeneous or heterogeneous matrix, to form small capsules. It can envelop a solid, liquid, or gaseous substance within another substance in a very small, sealed capsule [13].

In this study, the protein–polyphenol copolymer based on gallic acid–Antarctic krill peptides copolymer (GA-AKP) was used. Proteins and polyphenols are commonly found in food, and the protein group can serve as an effective carrier for polyphenol delivery, resulting in the formation of polymers with relevant functional properties [14]. Both Antarctic krill peptide and gallic acid have demonstrated hypoglycemic abilities [15], and their combination has been shown to improve hypoglycemic capacity. However, because of its environmental sensitivity, it must be protected and delivered in an appropriate form. To achieve this, a W_1_/O/W_2_ (water-in-oil-in-water) compound emulsion was used as the technical method. PLGA was used as the delivery vehicle, and GA-AKP was used as the delivery material. A high-pressure microjet method was employed, and a dual delivery system, GA-AKP nanocapsules (GA-AKP-Ns), was constructed. The effects of oil-phase PLGA concentration, jet pressure, and the number of jet cycle treatments on the particle size, PDI, zeta potential, rheological properties, centrifugal stability, and encapsulation rate of the microemulsions were investigated through single-factor experiments. Based on these findings, solid nanocapsules were obtained through lyophilization, and their physicochemical characteristics were characterized. Additionally, this study determined storage stability indexes, providing insights into the potential long-term viability of the GA-AKP-Ns dual delivery system. The use of nano-microcapsules and novel microjet methods in related areas will be expanded, promoting the preparation and utilization of these systems for the protection and delivery of active ingredients.

## 2. Materials and Methods

### 2.1. Materials

Polylactic acid–hydroxyacetic acid (PLGA, 50:50, Mw = 10,000–15,000, 34346-01-5) was purchased from Shanghai Peng Sheng Biotechnology Co. (Shanghai, China). Dichloromethane (DCM, 75-09-2) was purchased from Sinopharm Chemical Reagent Co. (Shanghai, China). Polyvinyl alcohol (PVA, 9002-89-5) was purchased from Johnson & Johnson Technology Co. (Shanghai, China) AbFluorTM 594 Labeling Kit was purchased from Thermo Fisher Scientific Co. (Shanghai, China) All other reagents are analytically pure.

### 2.2. Preparation of GA-AKP Microemulsions

GA-AKP microemulsions were prepared with the method described by Shuaikai [6], with slight modification. The internal aqueous phase was a GA-AKP copolymer solution. Copolymers were prepared, GA and AKP were dissolved in a phosphate buffer solution at pH7, and the final concentration of 1 mg/mL GA mother liquor and 10 mg/mL AKP mother liquor were formed, then 2.5 mL of GA mother liquor was added into 5 mL of AKP, and the volume was fixed to 10 mL. The mixed liquid was magnetically stirred at 200 r/min (25 ± 1 °C) for 5 min under the oxygen barrier to obtain the phosphate solution of the GA-AKP complex (internal aqueous phase) [16]. PLGA was dissolved in DCM as the oil phase (15, 20, and 25 mg/mL). PVA was dissolved in deionized water as the external aqueous phase (10% by mass). The ratio of the internal aqueous phase/external oil phase was 1:5, and a W_1_/O primary emulsion was formed by homogenization at 8000 rpm using a motorized homogenizer (D-130, Wigan Technology Co. (Beijing, China). Subsequently, the W_1_/O primary emulsion was added to the external aqueous phase, maintaining an internal oil phase to an external aqueous phase ratio of 5:25. The emulsion was then homogenized at 8000 rpm to form the W_1_/O/W_2_ primary emulsion. The secondary emulsification of the primary emulsion was carried out with a High-Pressure Microjet (SPCH-EP-IC-16-30, Shanghai Shengqi Instrument Technology Co. (Shanghai, China) under pressure conditions of 120, 150, 180 MPa, and 1, 2, 3 cycles, at 20 ± 0.5 °C, resulting in the formation of W_1_/O/W_2_ emulsions.

### 2.3. Preparation of GA-AKP-Ns

The W_1_/O/W_2_ compound emulsion was rotary evaporated under reduced pressure for 4 h to remove the organic solvent using a rotary evaporator (SN-RE-2000B, Shanghai, China), and then centrifuged at 10,000× *g* for 20 min to remove the supernatant using a high-speed refrigerated centrifuge (GL-20B, Shanghai, China). The remaining sample was lyophilized to obtain solid nanocapsules using a vacuum freeze-dryer (Marin Christ Company, Osterode, Germany). The entire process of micro-encapsulation is shown in Figure 1.

### 2.4. Characterization of GA-AKP Microemulsions

#### 2.4.1. Microemulsion Particle Size, PDI, Zeta Potential Determination

The particle size, polymer dispersity index (PDI), and zeta potential of the microemulsions were determined using a Zetasizer (Nano-ZS, Malvern Instruments, Worcestershire, UK) He/Ne laser with a scattering angle of 173° and a wavelength of 633 nm. The microemulsions were diluted 30 times with deionized water and measured in quartz cuvettes and electrode cuvettes, respectively. The temperature was set at 25 ± 0.1 °C.

#### 2.4.2. Centrifugal Stability

The stability of the obtained emulsion was investigated by centrifugation (GL-20B, Shanghai Anting Scientific Instrument Factory, Shanghai, China). The prepared emulsion was poured into a 25 mL centrifuge tube and then centrifuged at 5000× *g* for 5 min. The particle size of the emulsion before and after centrifugation was determined by the method described in Section 2.4.1.

#### 2.4.3. Microemulsion Viscosity Measurement

A rheometer (MCR301, Anton Paar Instruments Co., Shanghai, China) was used to determine the shear rheology tests of the emulsions. The diameter of the measuring parallel plate (PP 25) used was 25 mm, 1 mm in a gap, and the sweep scan range was from 0.1 to 100 s^−1^. The apparent viscosity of the sample was recorded by a shear sweep at 25 ± 0.1 °C.

#### 2.4.4. Encapsulation Rate of GA and AKP

Embedding rate: nanocapsules were subjected to lysis by adding a lysis solution (0.1 mol/L NaOH—2% sodium dodecyl sulfate (SDS) aqueous solution) and allowed to completely lyse at 37 ± 1 °C for 48 h to achieve optimal nanocapsule cleavage, then centrifuged at 10,000× *g* for 10 min to obtain supernatant and set aside for further analysis. Standard solutions ranging from 0.5 to 1.0 mg/mL GA and AKP were prepared, in which the AKP standard solution was dyed with biuret reagent. The standard curve of GA concentration was established at 270 nm (Y = 41.36X − 0.002, R^2^ = 0.9991, where Y is the UV absorbance of GA, X is the concentration of GA) and AKP at 540 nm (Y = 0.058X − 0.0014, R^2^ = 0.9986, where Y is the UV absorbance of AKP, X is the concentration of AKP). The embedding rate was calculated by the following formula:(1)$$\mathrm{Encapsulation}\;\mathrm{rate}\%=\left(\frac{m_{1}-m_{2}}{m_{0}}\right)\times 100\%$$
where m_1_ is the mass of GA/AKP in the supernatant after centrifugation, m_2_ is the mass of GA/AKP in the supernatant of microcapsules prepared from blank samples with phosphate buffer solution as core material, m_0_ is the input mass of GA/AKP.

### 2.5. Characterization of GA-AKP Nanocapsules

#### 2.5.1. Microscopic Morphology of Nanocapsules

The microstructure of nanocapsule particles was observed using scanning electron microscopy (S3400N, Hitachi Japan Co., Tokyo, Japan). A small amount of a sample was evenly spread on the surface of tin foil, pasted on double-sided conductive adhesive, and sprayed with gold, and we observed the microscopic morphology of nanocapsules at different magnifications.

#### 2.5.2. Internal Drug Distribution of Nanocapsules

GA-AKP was stained and labeled with an AbFluorTM 594 Labeling Kit, and microemulsions were prepared using the labeled material as the internal aqueous phase, as described in Section 2.2. The distribution of the material in the droplets was observed under a laser confocal microscope (TCS SP8, Leica Germany Corporation, Wetzlar, Germany), and processed by LAS AF Lite 2.6.0 software.

#### 2.5.3. Fourier Transform Infrared Spectrometry (FTIR) Analysis

Small amounts of GA-AKP, PLGA, and GA-AKP-Ns solid powders were scanned under FTIR spectra (Nicoletis-50, Thermo Fisher Scientific Co., Shanghai, China) in the range of 400–4000 cm^−1^.

#### 2.5.4. In Vitro Dissolution Experiment and Drug Release Kinetics of Nanocapsules

The cumulative release of GA-AKP in simulated gastrointestinal fluid before and after encapsulation was as follows [17].

The samples (50 mg) were incubated in centrifuge tubes with 2 mL of pH1.0 PBS and pH7.4 PBS as the solvent before and after embedding and shaken at 37 ± 1 °C, and the supernatants were removed by centrifugation. Throughout 12 h, the supernatants were taken every 2 h, and the tubes were supplemented with equal amounts of solvent, then the amounts of GA and AKP released were measured and calculated as in the method described in Section 2.4.4, respectively. The cumulative release rates of the samples were measured both before and after 264 h of embedding and fitted to the zero-order equation, the first-order equation, and the Higuchi equation to determine the optimal release model of hypoglycemic nanocapsules.

#### 2.5.5. Moisture Absorption, Thermal and Storage Stability of Nanocapsules

According to the table of relative humidity (RH) of saturated salt solutions, the RHs of each salt solution closed container at 10 ± 1 °C were 23%, 33%, 43%, 56%, 65%, and 80%, respectively; we used saturated solutions of CH_3_COOK, MgCl_2_, K2CO_3_, Mg(NO_3_)_2_, NaNO_2_, and (NH_4_)_2_SO_4_. The hygroscopic weight gains of GA-AKP and GA-AKP-Ns (500 mg) were measured under different RH storage conditions [16].

The thermal stability of the samples was determined using a Differential Scanning Calorimetry (DSC) instrument (TAQ2000, TA Instruments, Inc., New Castle, DE, USA). Lyophilized sample powder (5.0 mg) was taken and sealed in an aluminum plate, and a sealed empty aluminum plate was used as a blank control. The heating rate was set at 10 °C/h, and the heating range was 40–150 °C. The flow rate of nitrogen was 30 mL/min. The inhibition rates of α-amylase, α-glucosidase, and DPP-IV were determined at 0, 7, 14, 21, and 28 days by placing the samples before and after embedding in open, clean containers at −20, 10, and 40 °C, respectively. Determination was made of α-amylase’s inhibitory effect [18]. Briefly, 10 mL of 5 mg/mL inhibitor was reacted with 3 mL of α-amylase solution (10 mg/mL, 2000 U/g) for 5 min at 37 °C in a water bath, and 4 mL of starch solution (10 mg/mL) was added and kept warm in a water bath. Every 15 min, 1 mL of DNS reagent was added to 2 mL of the mixed solution, boiled in the water for 5 min at 100 ± 1 °C, and then cooled to 25 ± 1 °C. The coolant was volume-determined to 10 mL, and the absorbance value was measured at UV 540 nm for 1 h. The reaction was carried out at 37 °C for 5 min, and 4 mL of starch solution (10 mg/mL) was added.

The inhibitory effect on α-glucosidase was determined [16], 10 mL of 5 mg/mL inhibitor and 10 mL of α-glucosidase solution (0.02 mg/mL, 5000 U/g) were taken and reacted in a water bath at 37 ± 1 °C for 15 min, and the reaction was continued with the addition of 10 mL of 2.5 mmol/L p-Ntrophenyl-α-d-glucoside (pNPG) solution. The reaction was continued by adding 10 mL of 2.5 mmol/L p-Nitrophenyl-α-d-glucopyranose (pNPG) solution, and then 1 mL of the mixed solution was taken every 15 min. The mixed solution was cooled to 25 ± 1 °C, and the absorbance value was measured at UV 400 nm for 1 h. The inhibition rate for both is calculated as follows:(2)$$\mathrm{Inhibition}\;\mathrm{rate}\%=\left(1-\frac{A1-A2}{A3-A4}\right)\times 100\%$$
where A1 is the experimental group, A2 is the background control phosphate buffer solution substituted for the α-amylase/α-glucosidase inhibitory solution, and A3 is the blank group, phosphate buffer replacement inhibitor. A4 is the blank control, phosphate buffer replacement inhibitor, and α-amylase solution/α-glucosidase is considered as the blank control group.

The determination of the inhibitory effect on DPP-IV [16] was made using the process that 50 mL of DPP-IV and 5 mg/mL 10 mL of inhibitor were added to a 96-well enzyme plate, and the reaction was carried out at 37 ± 1 °C for 10 min. Then, 25 mL of the substrate was added to continue the reaction for 15 min, and the absorbance value was determined by fluorescence detection (excitation wavelength 360 nm, emission wavelength 460 nm). The inhibition rate was calculated as follows:(3)$$\mathrm{Inhibition}\;\mathrm{rate}\%=\left(1-\frac{A1-A2}{A3}\right)\times 100\%$$
where A1 is the experimental group, A2 is the blank group, phosphate buffer replacement inhibitor. A3 is the control group, and selegiline inhibitor is considered a positive control.

### 2.6. Data Processing

All measurements were repeated at least three times, and the data were analyzed by SPSS (IBM SPSS Statistics version 13.0). The results were expressed as mean ± standard deviation means and were compared using a one-way analysis of variance followed by Duncan’s multiple comparing tests. Differences between means were considered significant when (*p* < 0.05). Graph pad prism8 software was used to process the experimental data and plotting.

## 3. Results

### 3.1. Characterization of GA-AKP Microemulsions

#### 3.1.1. Effect of PLGA Mass Concentrations on Physicochemical Properties of Microemulsions

PLGA is a biomaterial that exhibits good biosolubility, efficient drug encapsulation, and excellent film-forming properties. Figure 2A,B present the particle size, PDI, and zeta potential at different PLGA concentrations. As the concentration of PLGA in the oil phase increased, the W_1_/O/W_2_ microemulsion showed a gradual increase in particle size and PDI and a gradual decrease in zeta potential. The concentration of PLGA had a significant effect on the particle size of the microemulsions. The average particle sizes were 246.1, 285.4, and 312.4 nm at different PLGA concentrations. As the concentration of PLGA increased, the droplet particle size distribution became wider with a tendency of agglomeration. The results of the study showed that as the PLGA concentration increased, the viscosity of the oil phase increased, and then the oil droplet size increased. This increase in droplet size negatively affected the emulsion, leading to an increase in microsphere particle size and wall thickness [6]. The wall-to-core ratio is a crucial factor affecting the properties of nanocapsules. Insufficient wall loading may not provide enough load to the core, while excessive wall loading may result in particle aggregation. The PDI and zeta potential values of 25 mg/mL PLGA were significantly different from the other two groups. This indicated a wider droplet size distribution and poorer particle size homogeneity under this condition. However, in the Chinese yam polysaccharide PLGA nanoparticle stabilized Pickering emulsion study at different PLGA concentrations (1–7 mg/mL), there was a decrease in particle size and PDI with increasing PLGA concentration. This may be due to the higher nanoparticle concentration leading to larger interfacial area coverage, limiting droplet aggregation and resulting in smaller emulsion particle size [19]. The observed differences in trends may be attributed to the varying ranges of PLGA concentrations utilized and the different preparation methods employed. Specifically, when PLGA concentrations ranging from 20 mg/mL to 60 mg/mL were combined with high-speed shear to prepare microcapsules, an increase in particle size was observed with increasing concentration [6].

The efficiency of nanocapsule preparation can be improved by separating the aqueous phase solvent in microemulsions by centrifugation. The centrifugal stability of microemulsions was evaluated by measuring the particle size of nanodroplets before and after centrifugation. Figure 2C shows a significant increase in the particle size of nanodroplets in the 15 mg/mL PLGA group after centrifugation. Under these conditions, PLGA did not provide sufficient load to the core material, causing the free material to regroup under the action of centrifugal force and form a core material with higher molecular weight. In contrast, the emulsions with 20 and 25 mg/mL PLGA showed no significant difference in particle size before and after centrifugation, indicating good centrifugal stability.

The rheological properties of an emulsion not only reflect its mechanical properties but also provide a reference for subsequent processing and handling [20]. By observing the effect of shear rate on the viscosity of microemulsions, we can illustrate their static flow properties. Figure 2D shows the rheological properties of microemulsions at different PLGA concentrations. The apparent viscosity of the W_1_/O/W_2_ microemulsions decreased with increasing shear rate, showing the typical shear thinning tendency of pseudoplastic non-Newtonian fluids (0.1–10 s^−1^). The apparent viscosity of microemulsions decreased as the mass concentration of PLGA in the oil phase increased, due to the larger particle size of emulsions at high concentrations of PLGA. However, smaller droplets have a larger specific surface area and tighter polymerization, which increases the shear resistance of the microemulsions. As a result, smaller oil droplets increase the viscosity of emulsions, contributing to their anticoagulation effect [21]. The low viscosity of the emulsion formed by PLGA at high concentrations relative to the emulsion formed by PLGA at low concentrations indicates that it is less stable [22].

Figure 2E shows the encapsulation rates of GA and AKP at different PLGA concentrations. The results indicate that a higher concentration of PLGA in the oil phase leads to better encapsulation of raw material within the experimentally selected range. Specifically, the encapsulation rate of GA increased from 67.7% (15 mg/mL PLGA) to 75.2% (25 mg/mL PLGA), while the encapsulation rates for AKP increased from 72.3% (15 mg/mL PLGA) to 88.5% (25 mg/mL PLGA). The improved encapsulation of AKP by microemulsions may be attributed to the poor water solubility of GA. GA can be partially dissolved in organic solutions and lost by evaporation with the solvent. The optimal embedding of materials was achieved at 25 mg/mL PLGA, but the loading capacity of PLGA was relatively low. Although 20 mg/mL PLGA had a slightly lower embedding rate than 25 mg/mL, there was no significant difference between the two, so the material utilization rate of 20 mg/mL PLGA was more satisfactory. Optimizing the encapsulation rate of the carrier through the process, under the fixed wall-to-core ratio, can provide better protection and higher bioavailability of the active ingredient. This is important for the application of microcapsules.

#### 3.1.2. Effect of Jet Pressure on Physicochemical Properties of Microemulsions

Dynamic High-Pressure Microjet (DHPM) treatment is a technique used to efficiently prepare fine-grained homogeneous microemulsions, improving their rheological properties, textural properties, and stability [23]. In this experiment, the FPG12800 Y-valve of the microjet meter was selected to treat the sample. The initial emulsion was drawn into the high-pressure chamber through the inlet and then entered the Y-valve at supersonic speed under a given pressure. As shown in Figure 3A,B, the microjet-treated emulsions outperformed the untreated primary emulsions in terms of particle size, PDI, and zeta potential. Among the three selected pressure conditions, the optimal values were observed at 150 MPa, with an average particle size of 285.1 nm, a PDI of 0.13, and a zeta potential value of 29.73 mV. At a jet pressure of 180 MPa, the emulsion particle size and PDI increased, and the absolute value of the zeta potential decreased. This was likely due to the “overprocessing phenomenon” that occurs at this pressure condition, where the high energy density causes the aggregation of dispersed particles [24]. The preparation of nanoemulsions of Eucommia seed oil using a high-pressure microjet resulted in similar findings. The particle size of the microjet-treated emulsions decreased, and the absolute value of the zeta potential of the emulsion increased as the jet pressure varied from 0 to 60 to 100 MPa. However, when the jet pressure increased to 140 MPa, the absolute value of the zeta potential decreased [2]. Perhaps the poor stability of PVA caused it to decompose into carbon monomers due to excessive energy. When the pressure of the high-pressure microjet was high, it broke the chemical bonds of the substance, causing it to disintegrate [25].

As shown in Figure 3C, the particle size of the primary emulsion without microjet treatment increased significantly after centrifugation. In contrast, the experimental group treated by the microjet displayed a minimal change in particle size before and after centrifugation. It is worth noting that the optimal stability under centrifugation was observed at 150 MPa. The primary emulsion had a wide particle size distribution, and the droplets were unevenly dispersed. Smaller droplets were deposited on larger droplets, a phenomenon known as “Oswald ripening” [26]. The microjet treatment further emulsified the emulsion, resulting in highly refined and uniformly distributed droplets. The droplets formed a tightly packed arrangement on the interfacial film, which increases their resistance to aggregation and enhances emulsion stability. This finding is consistent with the results regarding particle size, the PDI, and the zeta potential of microemulsions.

Figure 3D shows the variation in microemulsion viscosity with shear rate before and after microjet treatment, indicating that microjet treatment can improve microemulsion viscosity. Under the experimental conditions of 150 MPa, this process improved the droplet interaction effect and provided better mechanical strength to the emulsion. The viscosity of the fluid initially increased, then subsequently decreased as the pressure increased. The stability of the microemulsions was optimized.

In Figure 3E, the encapsulation rates of GA and AKP in the initial emulsion were only 51.68 and 54.37%, respectively. After three cycles of treatment at 120, 150, and 180 MPa, the encapsulation rates of GA and AKP increased to 66.54%, 71.50%, 63.27% and 74.30%, 85.20%, 80.60%, respectively. The loading of GA and AKP in PLGA increased from 10.3 and 108.7 mg/g to 14.3 and 170.4 mg/g at 150 MPa, which effectively improved the wall utilization. With increasing pressure, the molecular loading energy in the system became too high, accelerating Brownian motion and reducing emulsion stability, which affects the embedding effect. In addition, high pressure may have altered the peptide structure, causing peptide molecules to polymerize with each other [27], hindering binding with PLGA and reducing peptide embedding efficiency. In conclusion, the optimal choice was a jet pressure of 150 MPa.

#### 3.1.3. Effect of Jet Cycles on Physicochemical Properties of Microemulsions

The number of jet cycles of emulsions determines the number of collisions between droplets, which directly affects the emulsification effect. Figure 4A,B demonstrate that the particle sizes, PDI, and zeta potential were all significantly affected by the number of cycles, and were all improved compared to the initial emulsion before jet treatment. As the number of cycles increased, the emulsion showed a regular pattern of decreasing particle size, decreasing PDI, and increasing absolute zeta potential. After three cycles, the average particle size of the emulsion was 285 nm, the PDI was 0.13, and the absolute zeta potential was 29.73 mV. Maybe increasing the number of cycles provides more opportunities for droplets to collide. This would have disrupted the equilibrium of the emulsion, releasing smaller droplets to be rewrapped and forming a tighter interfacial film. As a result, the homogeneity and emulsification rate of the emulsion improved. This finding is consistent with some research on using DHPM to make emulsions, which also found that multiple forces produced in DHPM could lead to smaller particle sizes [2].

As shown in Figure 4C, the particle size of the emulsions treated for a single cycle increased significantly before and after centrifugation. However, there was no significant difference in the particle size of the emulsions treated for two and three cycles before and after centrifugation. The wider particle size distribution of single-cycle emulsions caused small particles with high energy to deposit more easily onto large particles with low energy, increasing the average particle size of the emulsions. At the same time, single-cycle emulsions had lower viscosity, and higher viscosity inhibits the droplet drop during centrifugation, preventing droplet aggregation and emulsion stratification. The particle size of the three-cycle emulsion was maintained at 300 nm before and after centrifugation, which demonstrated good centrifugal stability.

As shown in Figure 4D, the viscosity of the microemulsion increased progressively with the number of jet cycles. Previous research has demonstrated that the emulsion viscosity is determined by a combination of droplet interactions and the degree of aggregation when the mass concentration in the emulsion is certain [28]. Multiple cycles improved droplet refinement, resulting in uniform droplet size, smaller droplet molecular gaps, increased droplet number, and denser arrangement in the system. As a result, emulsions were less susceptible to external damage.

Figure 4E shows that increasing the number of cycles effectively improved the encapsulation rates of GA and AKP within the experimental range. The encapsulation rates of GA and AKP were 63.28%, 69.39%, 71.50%, 63.77%, 76.34%, and 85.2% after single-, two-, and three-cycle treatments, respectively. The emulsification rate was improved through multiple cycles, resulting in stable encapsulation of the free core in W_1_/O/W_2_ droplets. Additionally, the gradual curing of PVA prevented droplet repolymerization and diffusion of the core material. In conclusion, it was found that three cycles of microjet flow gave the best results.

### 3.2. Microscopic Morphology of Nanocapsules

The GA-AKP-Ns prepared with optimal process parameters of 20 mg/mL of PLGA concentration, 150 MPa jet pressure, and three jet times were characterized by scanning electron microscopy, shown in Figure 5. The lyophilized microemulsions maintained good sphericity, and the nanocapsules were uniform in size, ranging from 300–400 nm. The nanocapsules were slightly larger than the microemulsions due to droplet aggregation before lyophilization. The capsules were dispersed without mutual adhesion to each other, and their surfaces were smooth and dense, without any depressions or cracks. Compared with the traditional spray-drying process, the nanocapsules prepared by the compound emulsion solvent evaporation method can avoid cracking and collapsing on the surface of the capsules after cooling due to the difference in air pressure between the inside and outside of the capsules during the high-temperature drying process [28].

### 3.3. Laser Confocal Microscopy of Nanocapsules

The core-shell structure of the capsule and the distribution of raw materials inside were observed by laser confocal microscopy (Figure 6). The drug-loaded particles optimized by microjet flow were spherically dispersed with similar size, and the droplets were uniformly distributed in the outer aqueous phase, and the PVA curing process prevented the fusion of droplets in the inner phase. Figure 6 shows that the inner phase W_1_/O structure was satisfactory, and the water-soluble core material was fully wrapped in the oil phase. This uniform distribution in the oil phase promotes the continuity of active substance release, reduces drug dosage, and minimizes side effects. Small droplets of lutein-EAC were successfully encapsulated in the inner aqueous phase by ultrasonic treatment [29]. Similarly, FITC-stained BSA achieved good encapsulation with a uniform distribution in the (CA/NaCS) core [30], demonstrating successful material encapsulation. The wall thickness of the nanocapsule wall can be adjusted by manipulating the volume of the external water phase or injection parameters. Therefore, the microencapsulation process using the microjet-optimized compound emulsion solvent evaporation method can overcome the negative effects of high temperature on the active substance. This method also enables efficient, high-volume, and continuous production, while improving the utilization of packaging material and reducing waste. The dispersibility of nanocapsules makes them suitable for use as oral formulations or as suspensions for inoculation by injection.

### 3.4. FTIR Spectra of Nanocapsules

Figure 7 illustrates the changes in the characteristic bands of the functional groups and chemical bonds in the molecule after the formation of the capsule structure. The GA-AKP-Ns showed a broad absorption peak at around 3300 cm^−1^, which was generated by the O-H in the material by hydrogen bonding and came from the active hydroxyl group coupling at the end of the hydrophilic core and wall molecules. The absorption peak was enhanced at 2900 cm^−1^ and came from the C-H on the saturated carbon stretching. The strong absorption peak at 1750 cm^−1^ came from C=O vibrations, and the presence of an external aqueous phase PVA led to an increase in this peak. The absorption peak appearing at 700 cm^−1^ came from the C-H face outward bending of olefins or aromatics and was generated by interbenzyl ring substitution, which confirms the presence of GA-AKP in the capsule.

### 3.5. In Vitro Simulated Release of Nanocapsules

The membrane of nanocapsules provides a physical barrier for the embedded contents, which is crucial to solving reduced dissolution in the strong acid system of the stomach and delayed release in the intestinal environment [31]. Figure 8 shows the cumulative release rate of the ingredients in simulated gastrointestinal fluid. For unembedded GA-AKP in the simulated gastric environment for 2 h, the cumulative release rates of GA and AKP were 58.2% and 20.3%, respectively. The cumulative release rates of GA and AKP were 9.4% and 3.6% for GA-AKP-Ns in the simulated gastric environment for 2 h, indicating that the delivery of the active in the form of nanocapsules can effectively reduce the diffusion in the stomach and increase the percentage of drug reaching the intestine. When the drug was released in the stomach, the cumulative release rates of GA and AKP before and after encapsulation were 97.3% and 62.4%, 28.5% and 34.8%, respectively, within 12 h.

Further analysis of the release showed that the raw material GA had a fast release rate in the gastrointestinal environment (Figure 8A). GA is an acid-resistant substance with good solubility and stability under acidic conditions, which results in a large release in the gastric environment. The release rate decreased slightly after entering the intestine, and the cumulative release was close to 100% at 3 h. When delivered via nanocapsules, the release of GA was only 2.2% within 1 h into the gastric environment and 7.1% in the latter hour. This may be because some of the GA was dissolved in DCM and retained on the surface or interstitial space of PLGA after lyophilization and subsequently released in the stomach. The release rate remained stable upon entering the intestine and accelerated after 7 h in the intestine, indicating significant disintegration of the nanocapsules at this time, leading to faster diffusion of contents. The raw material AKP had a significant sudden release in the first 1 h, and the cumulative release no longer increased after 4 h in the intestine (Figure 8B). This indicated that the poor sensitivity and environmental stability of peptides with regard to gastric digestion leads to irreversible denaturation or clumping of the peptide, reducing molecular mobility. Further, upon entering the intestine, it is easily broken down into small-molecule amino acids by a variety of intestinal enzymes, losing its unique functional activity [32]. AKP delivered via nanocapsules had a very low amount of release in the stomach, and the release rate was stable, which was consistent with the release pattern of GA. It indicated that nanocapsules with PLGA as a carrier could provide protection for the active substance in the gastric environment and sustained, slow release in the intestine, which may be ideal for the in vivo delivery of loaded GA-AKP.

### 3.6. Nanocapsule GA-AKP Release Kinetic Equation

Mathematical and semi-empirical models such as zero-order equations, first-order equations, Higuchi, and second-order polynomial models have been used to fit and explain the drug release process under various experimental conditions [33]. To further investigate the in vitro drug release kinetics of GA-AKP-Ns, the release of GA and AKP in simulated gastric fluid at 1, 3, 5, 7, 9, and 11 days was measured and fitted to three mathematical drug release models: the zero-order equation, the first-order equation, and the Higuchi equation [34]. Among them, the zero-order equation can fit the release model of a controlled-release formulation, and the first-order equation and Higuchi equation can fit the release model of a sustained-release formulation (Figure 9). By comparing the R^2^ of the three models, it was determined that the release patterns of the two drugs in GA-AKP-Ns were more in line with the Higuchi approach, with R^2^ values of 0.7701 and 0.9624, respectively, indicating that the release of the drugs was driven by the concentration difference. This dual drug delivery system with a slow-release effect can effectively improve drug utilization and reduce the frequency of drug administration and damage to the gastrointestinal tract. Similarly, the use of PLGA-encapsulated siRNA to achieve in vivo targeting of siRNA for bone tissue governance, and their drug release model, was consistent with the Higuchi equation [35].

### 3.7. The Stability of the Nanocapsule

#### 3.7.1. Thermal Stability

The effect of the nanoencapsulation process on the thermal stability of the raw material GA-AKP was analyzed using DSC results (Figure 10A). The results indicate that the nanocapsule structure increased the thermal denaturation temperature of GA-AKP from 76.3 °C to 88.7 °C, thereby improving the thermal stability of the raw material. Previous experiments confirmed that the interaction between GA and AKP is driven by covalent forces. The mechanical forces generated during the injection process may facilitate further assembly of GA-AKP, resulting in stronger thermal stability of AKP. In addition, the high pressure may lead to the forced unfolding of the AKP structure and enhanced binding to the solvent, thus increasing the peak thermal denaturation. However, microjet treatment to unfold protein structures reduced the protein stability [36].

#### 3.7.2. Hygroscopic Stability

The hygroscopicity of the microcapsule powders was measured to gain insight into their physical stability when stored in different humidity atmospheres [37]. Figure 10B shows the weight gain of GA-AKP before and after encapsulation in different relative humidity environments for 5 days. The weight of GA-AKP increased by less than 10% in the range of 0–43% RH but increased rapidly when the RH was greater than 43%. At 80% RH, partial dissolution was observed. Under the same humidity conditions, the weight change of GA-AKP after encapsulation was consistently less than that of unencapsulated GA-AKP. This effect was more pronounced under low-humidity conditions, indicating that PLGA hydrophobic polymers effectively reduced the moisture absorption of the raw materials. With a further increase in humidity, GA-AKP-Ns combined with water molecules in large quantities, causing the PLGA to degrade and preventing it from providing a dense hydrophobic layer. This phenomenon can be improved by adjusting the ratio of polymers in PLGA [27]. In summary, the nanocapsule structure improves the hygroscopic stability of the raw material.

#### 3.7.3. Storage Stability of Nanocapsules

Compared to plant polyphenols, polypeptides are less stable and more susceptible to environmental factors. Their biological activity level may decrease during storage. Nanocapsules can preserve GA-AKP activity by preventing the exchange of oxygen and water between GA-AKP and the environment. The main inhibition rates examined in the experiment were α-amylase, α-glucosidase, and DPP-IV. α-Amylase and α-glucosidase are the main enzymes for the digestion of starch in the human body, and they can catalyze and promote the hydrolysis of the alpha-1.4 glycosidic bond, which ultimately makes large molecules of starch into small molecules of glucose, thus increasing the body’s blood glucose [38]. DPP-IV is a transmembrane glycoprotein that cleaves X-proline dipeptide from polypeptides. It is expressed in a variety of tissues and is readily released from cell membranes in response to stimuli such as insulin resistance and chronic inflammation, where it acts on glucagon-like peptide-1 (GLP-1) and glucose-dependent insulinotropic polypeptide (GIP) and disrupts their chain segments, rendering insulin incapable of regulating blood glucose [18]. Therefore, the inhibition of the related enzymes can slow down the rise of blood glucose, and thus assist in the treatment and alleviation of the disease. Figure 11 shows the changes in the amylase, glucosidase, and DPP-IV inhibitory activity with storage time (28 d) before and after encapsulation at −20, 10, and 40 °C. The results showed that the retention rate of the inhibitory effect of each group at different storage temperatures decreased with increasing storage time. The activity retention rate was always better than that of unembedded raw materials when compared at the same temperature and duration.

First, the α-amylase inhibition activity results showed that the raw materials’ activity loss was the fastest at 40 °C, while the nanocapsule technology could slow down the raw materials’ activity loss rate (Figure 11A). The temperature of −20 °C was identified as the optimal condition for preserving the activity of the raw materials in the nanocapsules under the selected experimental conditions. Low temperatures are beneficial for both inhibiting the denaturation rate of proteins and maintaining the structural stability of PLGA, thereby preventing its rapid degradation during storage. After 28 days of storage at −20 °C, 94.7% of the amylase inhibitory activity of GA-AKP-Ns was retained. Worku found that Moringa oleifera leaf extract nanocapsules inhibited the rise in blood glucose better than the untreated extract [39].

The inhibition activity of GA-AKP against glucosidase increased by 12.4% for nanocapsules stored at 10 °C and 8.3% for those stored at 40 °C, which is generally consistent with the pattern shown in α-amylase. It has been pointed out that PLGA undergoes non-homogeneous degradation at a temperature below 10 °C and vice versa for homogeneous degradation. This suggested that under low-temperature conditions, PLGA was unable to accommodate the volume of water molecules, thus limiting their access to the interior and exerting better protection to the core material.

The results showed that the storage temperature could affect the inhibitory activity of nanocapsules against DPP-IV (Figure 11C). Under the storage condition of 40 °C, there was an instability of the raw material, which led to the loss of the ability of amino acids bound to the active center of DPP-IV, and the inhibitory effect on DPP-IV was significantly reduced (*p* < 0.05). In high temperatures, only 57.0% of the inhibition rate was retained after 28 days, which was improved by PLGA loading (*p* < 0.05), and the retention of inhibition was about 73.4% after 28 days; the results showed a significant improvement.

## 4. Conclusions

In this study, based on the traditional compound emulsion solvent evaporation method, the compound emulsion was further treated with high-pressure microjet technology to achieve the homogenization of the droplet refinement and improve the embedding rate and drug loading of nanocapsules. The study found that the microjet flow improved particle collision, resulting in smaller emulsion particle size, improved stability, and better encapsulation of AKP and GA. Additionally, the concentration of wall PLGA also had a significant impact on emulsion properties. These results demonstrate the significant improvements in microemulsion properties following microjet treatment. Furthermore, the emulsions were transformed into nanocapsules using the solvent evaporation method. This method resulted in uniform encapsulation of GA-AKP in the inner aqueous phase of the nanocapsules, providing good stability, slow-release quality, and hypoglycemic function. The use of a high-pressure microjet can be an effective approach for preparing encapsulation systems that enhance functional properties, ensuring the slow release and protection of bioactive ingredients.

## Figures and Tables

**Figure 1 foods-13-01177-f001:**
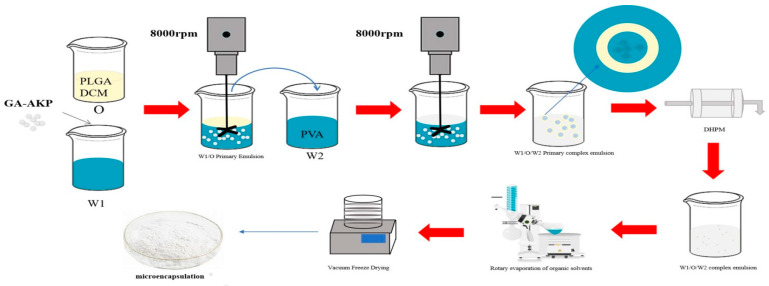
The preparation of the W_1_/O/W_2_ microencapsulation.

**Figure 2 foods-13-01177-f002:**
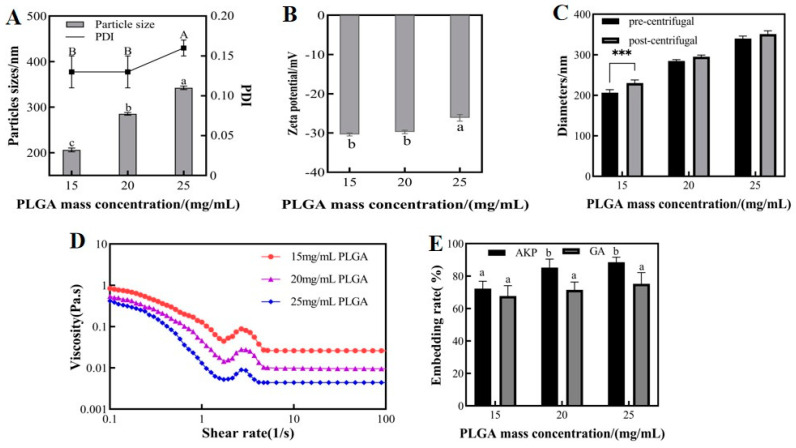
The particle size and PDI (**A**), zeta potential (**B**), centrifuge stability (**C**), viscosity properties (**D**), embedding rate (**E**) of microemulsion at different PLGA concentrations, 150 MPa jet pressure, and 3 jet cycles. Different upper and lower case letters indicate significant differences (*p* < 0.05) between groups. The *** indicates significant differences (*p* < 0.001) in particle size before and after centrifugation. Note: PDI: polymer dispersity index, PLGA: polylactic acid–hydroxyacetic acid, GA: gallic acid, AKP: Antarctic krill peptides.

**Figure 3 foods-13-01177-f003:**
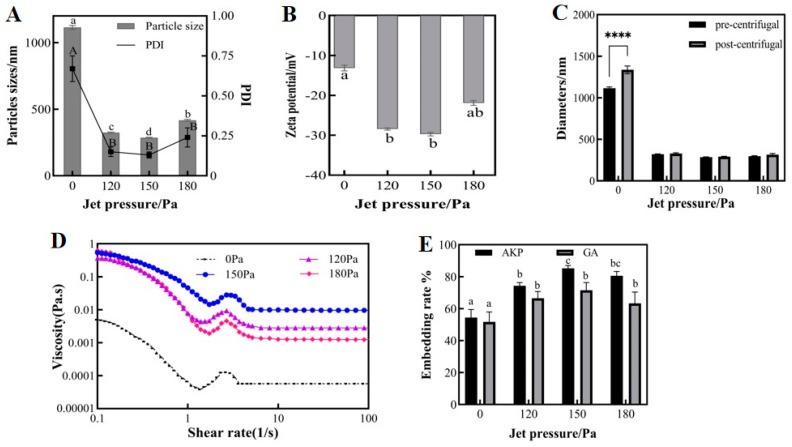
The particle size and PDI (**A**), zeta potential (**B**), centrifuge stability (**C**), viscosity properties (**D**), embedding rate (**E**) of microemulsion at different jet pressures, 20 mg/mL PLGA concentration, and 3 jet cycles. Different upper and lower case letters indicate significant differences (*p* < 0.05) between groups. The **** indicates significant differences (*p* < 0.0001) in particle size before and after centrifugation. Note: PDI: polymer dispersity index, PLGA: polylactic acid–hydroxyacetic acid, GA: gallic acid, AKP: Antarctic krill peptides.

**Figure 4 foods-13-01177-f004:**
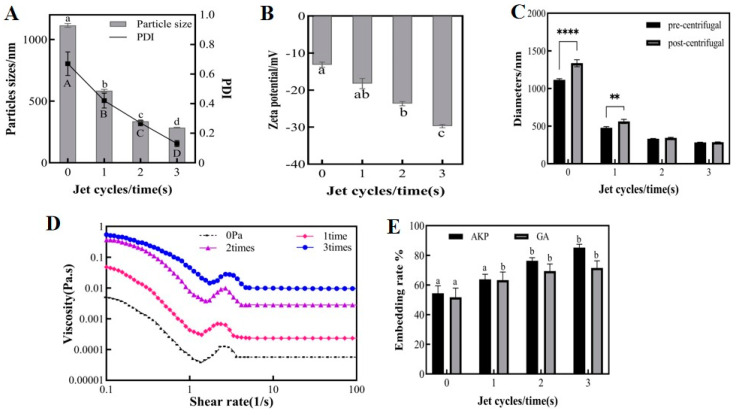
The particle size and PDI (**A**), zeta potential (**B**), centrifuge stability (**C**), viscosity properties (**D**), embedding rate (**E**) of microemulsion at different jet cycles, 20 mg/mL PLGA concentration and 150 MPa jet pressure. Different upper and lower case letters indicate significant differences (*p* < 0.05) between groups. The ** and **** indicate significant differences (*p* < 0.01) and (*p* < 0.0001) in particle size before and after centrifugation, respectively. Note: PDI: polymer dispersity index, PLGA: polylactic acid–hydroxyacetic acid, GA: gallic acid, AKP: Antarctic krill peptides.

**Figure 5 foods-13-01177-f005:**
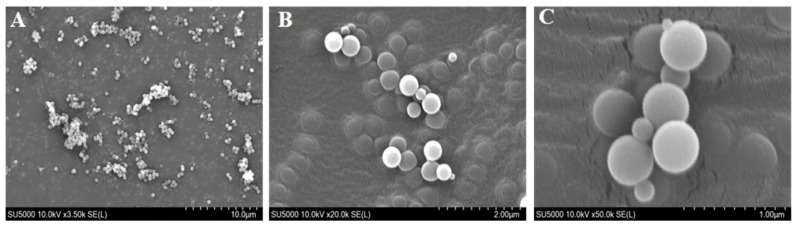
SEM micrographs of the GA-AKP-Ns under optimized conditions of 20 mg/mL PLGA concentration and 150 MPa jet pressure and 3 jet cycles ((**A**) 3500×, (**B**) 20,000×, (**C**) 50,000×). Note: SEM: scanning electron microscope, GA-AKP-Ns: gallic acid–Antarctic krill peptides nanocapsules, PLGA: polylactic acid–hydroxyacetic acid).

**Figure 6 foods-13-01177-f006:**
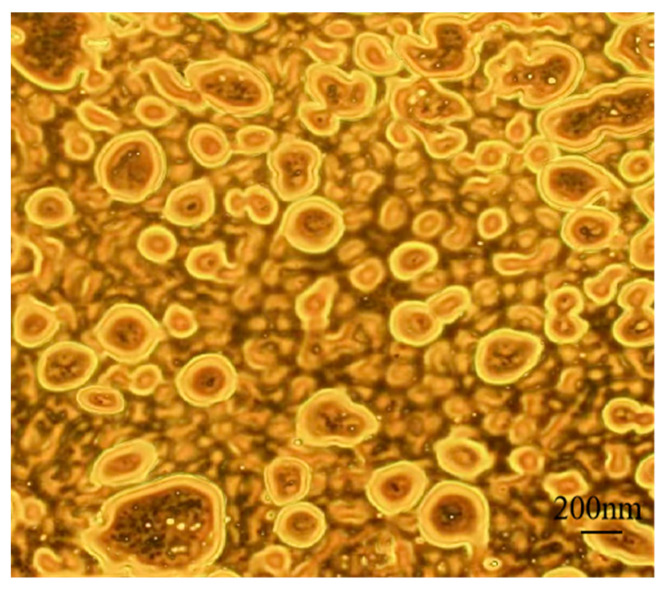
CLSM images of the GA-AKP-Ns under optimized conditions of 20 mg/mL PLGA concentration and 150 MPa jet pressure and 3 jet cycles. Note: CLSM: confocal laser scanning microscope, GA-AKP-Ns: gallic acid–Antarctic krill peptides nanocapsules, PLGA: polylactic acid–hydroxyacetic acid.

**Figure 7 foods-13-01177-f007:**
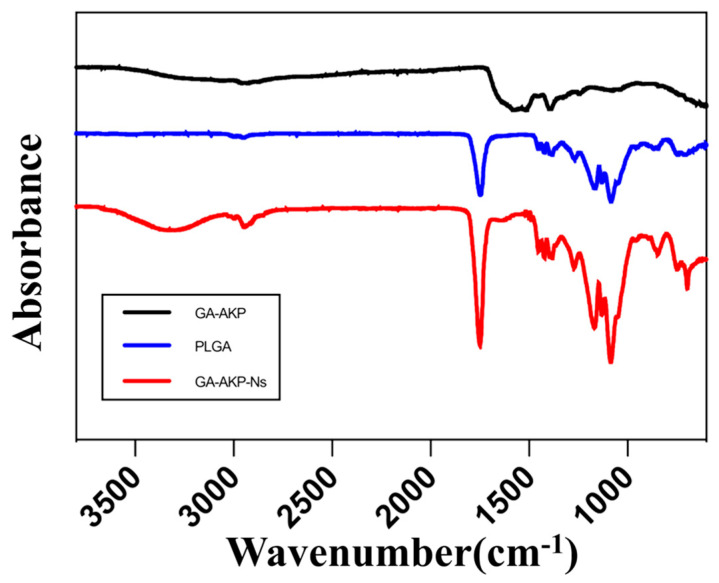
FTIR of GA-AKP, PLGA, and GA-AKP-Ns. Note: FTIR: Fourier transform infrared spectrometry, GA-AKP: gallic acid–Antarctic krill peptides copolymers, PLGA: polylactic acid–hydroxyacetic acid, GA-AKP-Ns: gallic acid–Antarctic krill peptides nanocapsules.

**Figure 8 foods-13-01177-f008:**
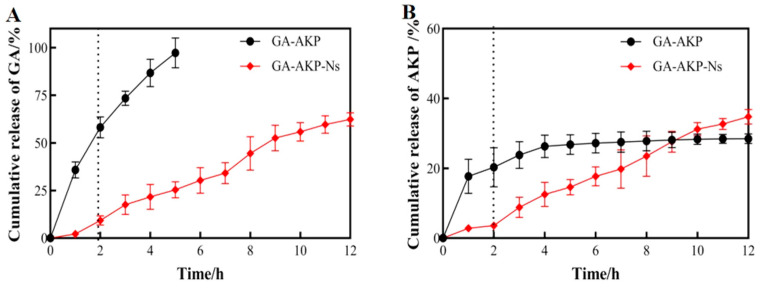
The cumulative release rate of GA-AKP and GA-AKP-Ns in gastrointestinal fluid ((**A**) GA release curves, (**B**) AKP release curves). Note: GA: gallic acid, AKP: Antarctic krill peptides, GA-AKP: gallic acid–Antarctic krill peptides copolymers, GA-AKP-Ns: gallic acid–Antarctic krill peptides nanocapsules.

**Figure 9 foods-13-01177-f009:**
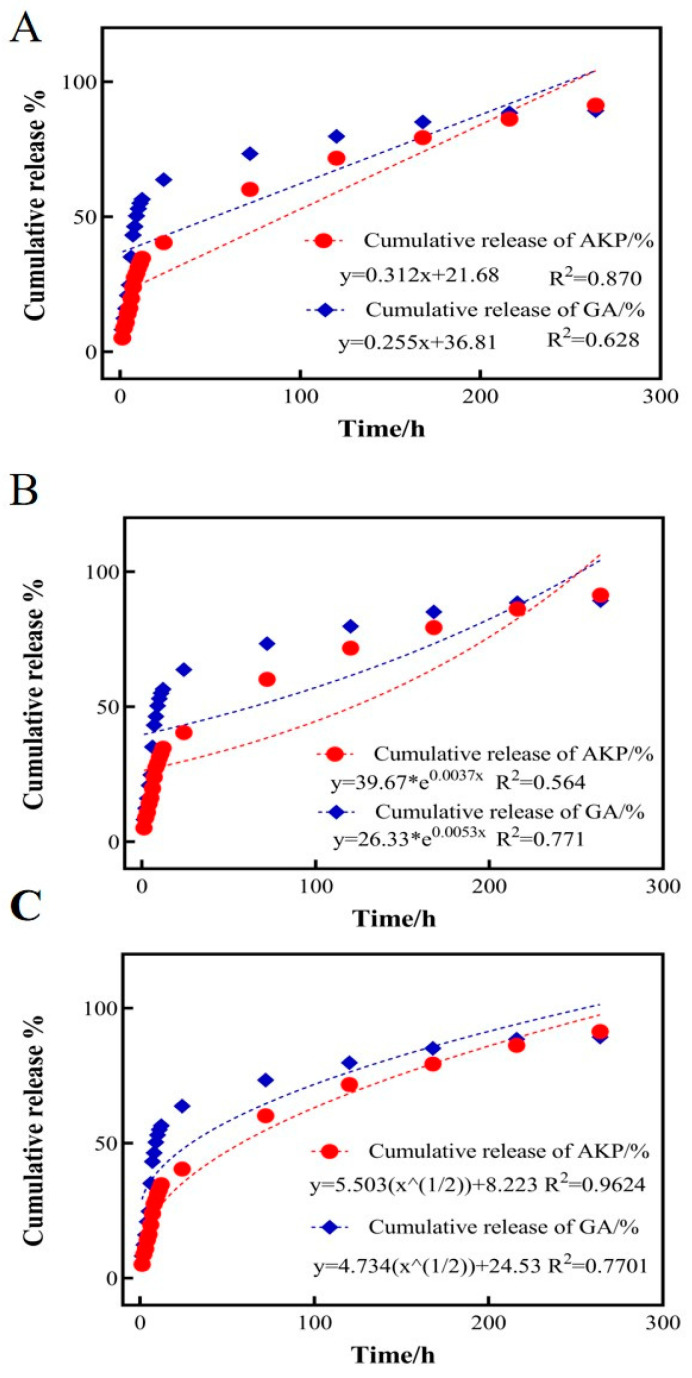
The GA and AKP release equation of GA-AKP-Ns, ((**A**) zero-order equation, (**B**) first-order equation, (**C**) Higuchi equation). Note: GA: gallic acid, AKP: Antarctic krill peptides, GA-AKP-Ns: gallic acid–Antarctic krill peptides nanocapsules.

**Figure 10 foods-13-01177-f010:**
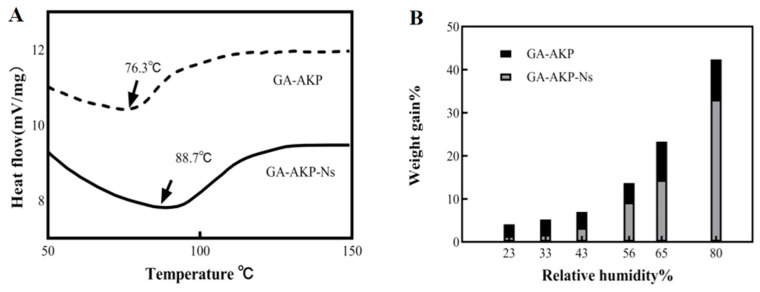
Thermal stability (**A**) and moisture absorption stability (**B**) of GA-AKP and GA-AKP-Ns. Note: GA-AKP: gallic acid–Antarctic krill peptides copolymers, GA-AKP-Ns: gallic acid–Antarctic krill peptides nanocapsules.

**Figure 11 foods-13-01177-f011:**
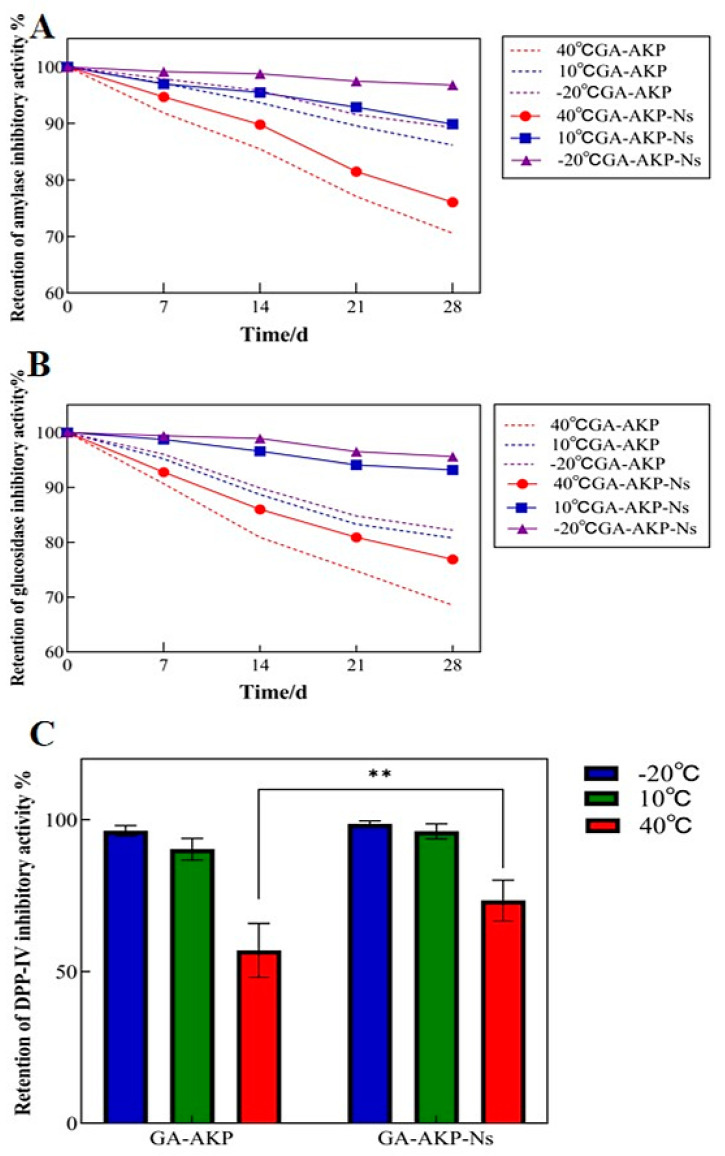
The inhibitory effect of GA-AKP and GA-AKP-Ns on α-amylase (**A**), α-glucosidase (**B**), and DPP-IV (**C**). Note: GA-AKP: gallic acid–Antarctic krill peptides copolymers, GA-AKP-Ns: gallic acid–Antarctic krill peptides nanocapsules. The ** indicates significant differences (*p* < 0.01) between samples at the same temperature.

## Data Availability

The original contributions presented in the study are included in the article, further inquiries can be directed to the corresponding author.

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
