# Peer review of "Preparation and Characterization of a Hypoglycemic Complex of Gallic Acid–Antarctic Krill Polypeptide Based on Polylactic Acid–Hydroxyacetic Acid (PLGA) and High-Pressure Microjet Microencapsulation"

_foods, 2024, doi:10.3390/foods13081177_

Round 1
Reviewer 1 Report
Comments and Suggestions for Authors
REVIEW OF MS TITLED: Preparation and characterization of a hypoglycemic complex of 2 gallic acid-Antarctic krill polypeptide based on PLGA and 3 high-pressure microjet microencapsulation
BY: Mengjie Li et al.
General comment: This manuscript gives a sound scientific account of the bioactive molecules loading into w/o/w doble emulsions. The appropriate use of wall materials and their concentration was the key to obtain a good loading of bioactive molecules into the nanoparticles. The use of a microjet equipment using high energy input into the particles achieved after a fixed number of cycles a stable emulsion, avoiding re-polymerization and diffusion of the core material. However, the particles size although they may behave as a good delivery system, showed a size that cannot be referred as “nanoparticles” because their diameter is >100 nm. Besides, the pressure used should have been well above the one reported, which needs to be corrected.
Therefore, the manuscript requires major corrections.
Abstract
I consider that the authors require to write more numeric results in the abstract to attract larger number of interested readers.
L. 23. It is not clear if the low pressure shown here is vacuum, absolute or manometric pressure. Please clarify.
L. 26 The acronym PLGA is misspelled. Please correct.
L. 28. The authors used in vitro tests for actual effectiveness and bioavailability of the bioactive peptide and GA, but in vivo experiments are required to be sure this system works as expected. Therefore, this is more a perspective than a conclusion of this work. Please correct.
General Comments
L. 85. It is not clear what the authors mean by “potential” and “rheological viscosity”. Please clarify.
L. 108-112. This paragraph is hard to understand. Please clarify.
L. 120. I consider that the 120 Pa pressure is too low, since it is almost atmospheric pressure. Please clarify.
L. 139. What was the probe of the rheometer, a cone and plate or a cylinder? Please clarify.
L. 145. The authors should use “x g”, instead of rpm. Please correct. What was the purpose of resting the prepared emulsion for 24 h, if it was going to be centrifuged? Please clarify.
L. 154-155. The “X” and “Y” values in the formulae need to be defined. Please correct.
L. 171-173. Sentences repeated. Please correct.
L. 187. The equations should be referred at “first order” and “second order” because they refer to the kinetics of release. Please correct
L. 271-272. The authors state that the apparent viscosity of nanoemulsions decreased due to smaller particle size, but in L. 275, they mention that the smaller droplets increased the viscosity of nanoemulsions. Please comment on this.
L. 258. It is not clear why the authors refer to the particles as nanoemulsions, since the size of a nanoparticle should be <100 nm. Please clarify, and correct throughout the text and figures.
L. 299. What is meant by “more satisfactory material utilization rate"? Please clarify
L. 308. Can the authors spell the acronym “DHPM”?. Please clarify.
L. 315-317. Bearing in mind that the equipment used imposes a high pressure to the treated emulsion, it is not possible that pressures between 120 Pa and 180 Pa were used. Please correct throughout the text and figures.
L. 324, 326, and throughout the text. I consider much better to write “zeta potential” instead of just “potential. Please correct.
Fig. 3. Can the authors explain why the zeta potential changed from (+) to (-) from the second to the third cycles? Please clarify.
Fig. 3a. The authors need to clarify what image correspond after one, two or three cycles throughout the microjet equipment. Please clarify.
Fig. 4. The authors should give in the caption of this figure an explanation of each image shown.
Fig. 5. The authors should explain what the reader is viewing for each image. Please clarify.
L. 505-506. It is not clear why the authors modeled the kinetics of drug release during in vitro gastric simulation up to 11 days, when It is known that the gastric phase of digestion takes no more than 6 h for healthy people. Please clarify.
L. 507. The equations developed for the kinetics of reaction are usually denoted as “zero order”, “first order”, “second order”, and so on, not levels. Please correct.
Figure 8. The caption of this figure is misplaced. In addition, it does not explain what phase of simulated gastrointestinal digestion is being modelled. Please clarify.
L. 526-527. The authors state that covalent forces are present in the interaction between GA and AKP, but they do not prove this assumption. For instance, the FT-IR spectroscopy does not show evidence of covalent bonds formation. Please clarify.
L. 560. It is not clear what the authors mean by enteroglycans, since DPP-IV hydrolyses peptides, not glycans. Please clarify.
L. 579-581. This paragraph is confusing and needs re-writing, because it seems that alfa amylase inhibition activity increases with temperature. Please clarify.
L. 600-603. This paragraph is a repeat of the results, and is redundant. Please delete.
References. The initials of all authors should be written after their surname, and the journals name should be abbreviated. Please correct.
Comments on the Quality of English Language
L. 85, ... the microstructure, particle size, PDI, potential, rheological viscosity, L.. 108-112. The GA mother liquor was added into 5 mL of AKP, and the final molar ratio of the two 107 was 1:1, and the volume was fixed to 10 mL, and the volume was fixed to 10 mL with 108 oxygen-insulated magnetic stirring for 5 min under the condition of room temperature, 109 and the phosphate solution of GA-AKP complex was obtained under the condition of ox-110 ygen-insulated magnetic stirring for 5 min under the condition of room temperature (in-111 ternal aqueous phase) [15].
L. 579-581. The inhibition activity of GA-AKP against glucosidase increased by 12.4% for 579 nanocapsules stored at 10 and 8.3% for those stored at 40 ℃, which is generally consistent 580 with the pattern shown in α-amylase (Figure. 9d).
Reviewer 2 Report
Comments and Suggestions for Authors
When mentioning an abbreviation for the first time, it is necessary to decipher it.
Line 80 - What is "W/O/W"?
Line 109 - Need to be more specific about the temperature and what was the power of the magnetic stirrer (how many revolutions per minute)?
Line 114, 119, 123, 124 and other - Exactly what equipment was used?
Line 136 and other - Was it really possible to ensure such an accurate temperature throughout the experiment? Usually the temperature is indicated with a possible offset of "±"!
Line 151-152 - More precise centrifugation parameters should be specified.
Line 186 - Was it really done "the samples were measured both before and after 264 hours of embedding"?
Line 206 and 216 - It should be specified more precisely at what temperature it happened.
I would recommend supplementing Section 3 with data from the scientific literature, comparing the obtained results or trends with the findings of other researchers.
Figure 1a), Figure 2a) and Figure 3a) - These all 3 images are already not the same, so I would recommend adding a), b) and c) to each image.
I would recommend adding the abbreviations used and their transcripts to each image. The image must be understandable even without the main text.
Line 308 - What is " DHPM "? Transcript!
Figure 4 and Figure 5 - I would suggest adding a), b), etc. and decipher and explain what is depicted in each.
Figure 7 - Design of graphics - the same font and letter size must be used.
Figure 8 - Graphics design - the letters are too small and it is not legible.
Line 563 - Is "Figure. 9 c-e ..." really meant here?
Reviewer 3 Report
Comments and Suggestions for Authors
The submitted paper (article) is focused on the preparation of Gallic acid-Antarctic krill peptides (GA-AKP) nanocapsules (GA-AKP-Ns) by emulsification technique based on microjet high pressure treatments. GA-AKP have used due to their functional properties (hypoglycemic ability), PLGA (in oil phase) is used as delivery vehicle. The effects of oil-phase PLGA concentration, jet pressure, and number of jet cycle treatments on the emulsified structure, particle size, PDI, potential, rheological behavior (viscosity), stability, and encapsulation rate of the nanoemulsions were investigated changing a single factor for each experiment.
The work done is solid and interesting, well organized and every aspect of produced emulsions-particles characterization has been evaluated. References appear all adequate. The discussion of achieved results (role of PLGA concentration, jet pressure intensity, number of jet cycle) on final products characteristics (size, load, stability- thermal, hygroscopic, after progressive storage - release properties - at physiological pH values - also in comparison to “naked” materials) seem reasonable, detailed and coherent with experimental observations.
However, I had some difficult to read and to fully understand some parts. Fundamentally the use of micro and nano words in several points do not help to understand why often products are called microcapsules (lines 27, 302, … 465, 533, 609, 610). Please clarify, also title and abstract should be revised (final part).
Final products, lyophilized particles that encapsulate GA AKP in PLGA wall, are discussed in terms of size as: “nanocapsules are slightly larger than nanoemulsions”. Please clarify the sense of line 609: “emulsions were transformed in microcapsules by solvent augmentation method”.
Preparative methodology could be presented by blocks flow diagram or schematized representation to help the comprension of the different step.
Some parts of Figures 1, 2, 3 are not well legible (b c e f) please adjust.
Figure 4 and Figure 5 should report the “optimized conditions” in their captions.
Figure 8 and Figure 9 are not well legible.
Minor notes:
The paper must be carefully revised for minor adjusts along the manuscript. In the following some indications as examples:
- ever acronym should be introduced as as extended test for before its use (line 80 WOW)
- CAS references of materials could be useful for Readers
- Correct PH in pH (lines 96-100)
Round 2
Reviewer 1 Report
Comments and Suggestions for Authors
I have no further comments to the authors. Corrections have been made throughout the mansucript
Comments on the Quality of English LanguageThere are few minor error in English Language, that may not affect the quality of the manuscript.
Reviewer 2 Report
Comments and Suggestions for Authors
Still review Fig.2, Fig.3 and Fig.4 and add transcripts of abbreviations used.
Still review all the places where temperatures are listed. Was it really possible to ensure such an accurate temperature throughout the experiment? Usually the temperature is indicated with a possible offset of "±"!
